# Non-Destructive Assessment of the Dynamic Elasticity Modulus of *Eucalyptus nitens* Timber Boards

**DOI:** 10.3390/ma14020269

**Published:** 2021-01-07

**Authors:** Alexander Opazo-Vega, Víctor Rosales-Garcés, Claudio Oyarzo-Vera

**Affiliations:** 1Department of Civil and Environmental Engineering, Universidad del Bío-Bío, Concepción 4081112, Chile; 2Department of Civil Engineering, Universidad Católica de la Santísima Concepción, Concepción 4090541, Chile; coyarzov@ucsc.cl; 3Department of Construction, Universidad del Bío-Bío, Concepción 4081112, Chile; vrosaleg@ubiobio.cl

**Keywords:** hardwoods, operational modal analysis, model updating, regional sensitivity analysis

## Abstract

*Eucalyptus nitens* is a fast-growing wood species with a relevant presence in countries like Australia and Chile. The sustainable construction goals have driven the search of structural applications for *Eucalyptus nitens*; however, this process has been complicated due to the defects usually presented in these timber boards. This study aims to evaluate the dynamic elasticity modulus (*E_xd_*) of *Eucalyptus nitens* timber boards through non-destructive vibration-based tests. Thirty-six timber boards with different levels of knots and cracks were instrumented and tested in a simply supported condition by measuring longitudinal and transverse vibrations. In the first stage, the *E_xd_* was calculated globally through simplified normative formulas. Then, in a second stage, the local variability of the *E_xd_* was estimated using operational modal analysis (OMA), finite element numerical simulations (FEM), and regional sensitivity analysis (RSA). The positive correlation found between the global static modulus of elasticity and *E_xd_* suggests that non-destructive techniques could be used as a reliable and fast alternative for the assessment of bending stiffness. Finally, the proposed method to estimate the local variability of *E_xdt_* based on the combination of OMA, FEM, and RSA techniques was useful to improve the structural selection process of timber boards for lightweight social housing floors.

## 1. Introduction

In recent years, there has been a growing interest in developing countries to use materials that promote sustainable construction. Among the materials used to achieve this goal, timber stands out because it has positive environmental attributes, including low embodied energy and low carbon impact [1]. Chile is among the 10 countries with the largest areas of planted forest [2], reaching a total of 22,900 km^2^ accumulated for the year 2017 [3]. The planted species with the highest potential for construction uses are *Pinus radiata* and *Eucalyptus nitens*. Traditionally, *Pinus radiata* has been destined in a higher percentage to generate timber for structural use. In contrast, *Eucalyptus nitens* has been more focused on the production of pulp, paper, and biofuels. However, in recent years, this trend has been trying to be reversed, because *Eucalyptus nitens* grown in Chile generally has better physical and mechanical properties than *Pinus radiata* [4].

Despite the potential of *Eucalyptus nitens* timber to be used as a structural element, its application in Chile is still incipient. On the one hand, the tensions induced by the tree growth processes and the subsequent drying process of the sawn timber affect the quality of its mechanical properties [5,6]. On the other hand, since Chilean *Eucalyptus nitens* forests are mainly managed for the pulp and paper industry, tree pruning or thinning processes are not usual, since the aim is to saw the wood in short rotation periods. Due to the above, *Eucalyptus nitens* sawn timber generally has significant variability in its mechanical properties, both inside a timber board and between boards [4,7,8]. Therefore, it is common to find many defects in this type of timber, such as knots, fiber deangulations, and cracks. All these defects frequently make *Eucalyptus nitens* unsuitable for structural use.

The search for structural applications for *Eucalyptus nitens* wood is a challenge that has been posed not only in Chile, but also in developed countries such as Australia. Among the reasons for the above, the high availability of this forest resource stands out, as shown by the 2700 and 2360 km^2^ in Chile [3] and Australia [9], respectively. In 2017, there was already concern about whether it was feasible to achieve applications in the construction industry with low structural grade *Eucalyptus nitens* boards [10]. One of the leading research lines has been the characterization of the physical-mechanical properties of individual *Eucalyptus nitens* boards. Derikvand et al. [11] evaluated the modulus of elasticity and modulus of rupture of 55 *Eucalyptus nitens* boards through four-point bending tests. With their results, they were able to identify which defects in the timber boards had the most significant influence on their mechanical properties. Later, the same research team extended their results to other mechanical properties, such as shear and compression strength in the parallel and perpendicular directions to the wood fiber, for small-sized specimens [12].

Seeking to improve the mechanical properties of this type of timber, Wentzel et al. [13,14] studied the feasibility of increasing the physical-mechanical properties of *Eucalyptus nitens* boards through thermal modification, looking for its application in outdoor exposed decking. Another novel research line has been to the production on structural panels made of *Eucalyptus nitens*. Among the various applications, plywood [15], LVL [16], cross-laminated timber [17], and nail-laminated timber [18,19] stand out. The results of these studies show that the selection of timber boards with optimal structural quality is crucial for generating high standard panels.

One of the most used indicators of structural quality in timber elements is the modulus of elasticity parallel to the grain (*E_x_*). *E_x_* is an elastic property related to the element’s stiffness and, therefore, associated to deformations under service loads. Consequently, an accurate estimation of this elastic property is crucial because the structural design of timber constructions is mostly controlled by serviceability requirements than by its strength. The traditional way to calculate *E_x_* of a timber board is through a four-point pseudo-static bending test, performed in certified laboratories [20]. However, in recent years, several researchers have looked for faster, non-destructive techniques based on short-duration dynamic excitations [21,22,23,24,25,26,27,28,29]. The so-called dynamic modulus of elasticity (*E_xd_*) determined by this vibration technics is, then, correlated to the *E_x_* obtained by applying the pseudo static bending test.

The main differences between these non-destructive techniques were the source and the direction of the dynamic excitation. Some research has focused on applying the dynamic excitations in the longitudinal direction of the timber boards [21,22,23,24,25]. Among the advantages of these techniques, their application speed stands out, since sensors are only located at the ends of the timber boards, without the need for special support conditions. In contrast, in the investigations developed in [26,27,28,29], dynamic excitations were applied in the transverse direction of timber boards. These techniques are an excellent alternative to estimate *E_xd_* when it is impossible to access the ends of the wooden boards (for example, for in-situ field testing), or when its dynamic properties need to be known for numerical model calibration. These studies demonstrated that regardless of the direction of the used dynamic excitation, it is possible to find a positive correlation between *E_x_* and *E_xd_*, with correlation coefficients that range between 0.90 and 0.97. Therefore, an indirect way to accurately calculate *E_x_* of a timber board is by estimating its *E_xd_* through these kinds of non-destructive techniques.

Previous work associated with the estimation of the *E_xd_* for timber boards has not included the *Eucalyptus nitens* species. Furthermore, research has focused only on estimating the *E_xd_* globally, i.e., an average value in a timber board, without analyzing the variability of this quality indicator along of the structural element. An analysis of the local variation of *E_xd_* in different areas of a timber element can be very relevant when evaluating wood species with many defects, such as *Eucalyptus nitens*. The aim of our work is to estimate the variability of *E_xd_* in *Eucalyptus nitens* boards through non-destructive testing based on vibration response. The timber boards come from fast-growing Chilean forests, so there is a significant variation in the amount and distribution of wood defects. With the results of the present research, it is expected to generate new indicators for the structural quality control of *Eucalyptus nitens* timber boards, which are applicable both in sawmills and in on-site constructions. In this way, the use of this type of timber in buildings will be enhanced, promoting a more sustainable construction industry.

## 2. Materials and Methods

This study was divided into two major stages. In the first stage, the timber boards are subjected to longitudinal and transversal vibration tests to estimate the global average *E_xd_* and find possible correlations with *E_x_* values obtained in traditional pseudo-static bending tests. In a second stage, a methodology is proposed to study the variability of *E_xd_* within the timber elements. This methodology combines the transverse vibration test results with Operational Modal Analysis (OMA), finite element model updating techniques (FEMU), and regional sensitivity analysis (RSA).

### 2.1. Description of the Timber Boards

A sample of 36 *Eucalyptus nitens* boards was selected to perform a series of non-destructive tests. The boards came from a 19-year-old tree plantation located in the Ñuble Region of Chile, with an average diameter of 40 cm. The initial purpose of these forests was the production of wood for pulp and paper industry, so they were not subjected to thinning or pruning processes. Therefore, the selected timber boards had many defects, such as knots, cracks, checks, fiber deangulations, pith, and the presence of juvenile wood, among others. Besides, the timber boards came from different parts of the tree trunks. Thirty-three percent of the wood boards had annual ring angles between 0° and 15° (i.e., back-sawn boards), 39% between 30° and 60° (i.e., transitional-sawn boards), and 28% between 75° and 90° (quarter-sawn boards). Figure 1 shows the central third and the transversal section of the timber boards #33, #28, and #10, which presents different amounts of defects.

The timber boards had very different moisture contents, ranging from 13.40% to 26.10%. Therefore, to decrease the influence of this variable on subsequent results, the boards were placed in a climate chamber for 14 days, with controlled temperature (20 °C) and relative humidity (65%). After taking them out of the climate chamber, their moisture content and density were measured before performing the static and dynamic non-destructive tests. This second measurement was done to verify that all timber boards had less than 20% moisture content. This limit is required by Chilean regulations for wood used in the construction industry. The moisture content was measured with a Wagner dielectric xylohygrometer, model L612, averaging at least three measurements across each board. For calculating density, the mass of each board was measured on a calibrated balance and then divided by volume. This volume was calculated with the average dimensions measured in at least three sectors of the timber boards. Table 1 summarizes the dimensions, visual characteristics, and physical properties of the timber boards.

Of all the parameters indicated in Table 1, the one that had the most significant variability was the continuous clear wood length, with a coefficient of variation close to 60%. This parameter represents the maximum defect-free length found in a timber board. Therefore, the lower the value of this parameter, the greater the board’s presence of defects. Based on the above, it is expected that this parameter will be one of the most influential in subsequent evaluations.

### 2.2. Non-Destructive Tests for E_x_ and E_xd_ Global Estimation

Three types of non-destructive tests were performed on the wooden boards: Longitudinal vibrations, transverse vibrations, and static bending. The purpose of the first two test sets was to estimate the *E_xd_* through two different techniques and then try to correlate those values with their corresponding *E_x_* obtained with the last set of static bending tests. The following sub-sections describe the laboratory implementation of each of these tests.

#### 2.2.1. Longitudinal Vibration Tests

These tests were conducted with a Timber Grader MTG device developed by Brookhuis Applied Data Intelligence. This handheld device incorporates both an activator and a stress wave detector, so only one end of the timber board needs to be accessed. The device’s principle is to generate a stress wave that travels to the end of the board in its longitudinal direction and then reverberates and returns to its point of origin. The device detects the stress wave’s resonant frequency and estimates *E_xd_* value using Equation (1) [21].
(1)$$E_{xdl}=4\cdot \rho \cdot L^{2}\cdot {\left(f_{L1}\right)}^{2}$$
where $E_{xdl}$ is the dynamic modulus of elasticity determined by longitudinal vibration, ρ is the timber board density (kg/m^3^), L is the timber board length (m), and $f_{L1}$ is the first longitudinal stress wave resonant frequency (Hz). Finally, the results are transmitted to a laptop computer via Bluetooth. All timber boards were supported at their ends in 2 points with a flatwise orientation. Figure 2 shows the laboratory implementation of the longitudinal vibration test.

#### 2.2.2. Transverse Vibration Tests

The transverse vibration tests were performed following the recommendations of the ASTM D6874 standard [30]. The main idea of this test is to arrange the timber boards in a simple-supported condition at their ends and then excite them through low energy impacts in the vertical direction (perpendicular to its longitudinal axis). The transversal vibrations generated by the impact are registered through one or more sensors distributed along the timber boards. Then the measured signals are processed to identify the first transverse resonant frequency. Finally, *E_xd_* can be estimated with Equation (2).
(2)$$E_{xdt}=\frac{{\left(f_{T1}\right)}^{2}\cdot w\cdot {\left(L_{s}\right)}^{3}}{K_{d}\cdot I\cdot g},$$
where $E_{xdt}$ is the modulus of elasticity determined by transverse vibration, $f_{T1}$ is the first transverse resonant frequency (Hz), w is the weight of specimen (N), $L_{s}$ is the span length equal to  0.98L (mm), $K_{d}$ is a constant for free vibration of a simply supported beam (2.47), I is the specimen moment of inertia (mm^4^), and g is the acceleration due to gravity (9807 mm/s^2^).

In order to obtain $f_{T1}$, the ASTM D6874 standard indicates that it is necessary to install only one sensor that records the vertical vibrations at the middle of the supported length of the timber boards. This experimental configuration is suitable to obtain a global value of *E_xdt_* using Equation (2). However, the transverse vibration method also allows to determine additional dynamic properties, such as higher vibration frequencies with their respective damping ratios and modal shapes, by employing more sensors. These additional dynamic properties can be used to estimate the local variation of *E_xdt_* due to the different types of wood defects. Therefore, to achieve this last objective, vertical accelerations were recorded by five uniaxial accelerometers, evenly spaced, and attached to the bottom face of timber boards (i.e., 1/6th, 2/6th, 3/6th, 4/6th, and 5/5th of span length). Integrated circuit piezoelectric (ICP) accelerometers (model 603C01, IMI Sensors, Depew, NY, USA) were used, with a sensitivity of 100 mV/g. The data acquisition system consisted of a multi-channel dynamic signal acquisition module (model NI 9234) gathered into a Compact DAQ chassis (model cDAQ-9174, National Instruments, Austin, TX, USA) and linked via USB to a laptop. Data were collected at a sampling rate of 1652 Hz, and the timber boards were in an edgewise orientation. The typical accelerometer locations and instrumentation setup are shown in Figure 3.

#### 2.2.3. Static Bending Tests

The last set of non-destructive tests corresponded to the execution of traditional static bending tests following the EN408 standard [20]. Besides, all timber boards were supported at their ends in 2 points with an edgewise orientation. The typical configuration of these tests is shown in Figure 4, and the expression that allows calculating *E_x_* is indicated in Equation (3).
(3)$$E_{x}=\frac{3\cdot a\cdot l^{2}-4\cdot a^{3}}{2\cdot b\cdot h^{3}\cdot \left(2\cdot \frac{w_{2}-w_{1}}{F_{2}-F_{1}}-\frac{6\cdot a}{5\cdot G\cdot b\cdot h}\right)},$$
where b and h are the thickness and the width of the board (mm), l is the span length equal to 18h±3h (mm), a is equal to 6h±1.5h (mm), $F_{2}-F_{1}$ is an increment of load (N) in the linear-elastic range, $w_{2}-w_{1}$ is the increment of vertical displacement (mm) corresponding to $F_{2}-F_{1}$, and G is the shear modulus of the board.

### 2.3. Methodology for Estimating the Local Variation of E_xdt_

As indicated in Section 2.2.2, the transverse vibration test was modified to estimate the local variation of *E_xdt_* in the presence of various defects in the wood. One useful technique to face this challenge is the finite-element model updating, also known as a model calibration or parameter estimation [31]. This technique allows the reconstruction of some unknown properties of a system, which appear as parameters in a numerical model, from the observation of other system properties using experimental tests. Based on the above, a three-stage methodology was implemented, as shown in Figure 5.

In general terms, the timber boards’ dynamic properties are firstly estimated using Operational Modal Analysis (OMA) techniques. Then, a series of numerical simulations are executed from finite element models of the wood panels. Finally, the numerical models that best fit the experimental data are selected through Regional Sensitivity Analysis (RSA). The three stages mentioned above are described in detail below.

#### 2.3.1. Operational Modal Analysis

Previous research [32] has shown that Operational Modal Analysis (OMA) is useful for estimating the dynamic properties of structural timber elements from vibration response measurements only, both in laboratory and in-situ construction contexts. Over the years, several OMA methods have been developed, classified as parametric and non-parametric. One of the advantages of parametric methods is that they generally show better performance than non-parametric ones. However, they have the disadvantage of using very complicated mathematical procedures with a high computational cost. Conversely, non-parametric methods are quicker and more manageable to utilize in field tests since they permit a prompt view of the measurements’ effectiveness and the dynamic identification results [33]. Consequently, it is valuable to apply both parametric and non-parametric methods to achieve a more reliable identification of a structural element’s dynamic properties.

Two broadly accepted OMA methods were chosen: Enhanced Frequency Domain Decomposition (EFDD) [34,35] and Stochastic Subspace Identification (SSI) [36,37]. EFDD is based on the frequency-domain (non-parametric) approach, whereas SSI is based on the time-domain (parametric) approach. With these methods, it was possible to identify the vibration frequencies, damping ratios, and modal shapes of the timber boards using ARTeMIS Modal Pro software [38].

EFDD allows identifying even closely spaced modes through the singular value decomposition (SVD) of the power spectral density matrix (PSD). Some necessary tests and data processing rules to obtain reliable dynamic property estimations with the EFDD method have been suggested by Rainieri et al. [39]. Firstly, it is recommended to get a total vibration record length equal to 1000–2000 times the first natural structural period to maintain low PSD estimation errors. Some preliminary numerical modal analysis results showed that the most flexible timber boards had the first vibration period close to 0.03 s. Hence, a length of 60 s was set for all the vibration records. Secondly, selecting an appropriate frequency resolution is necessary because some dynamic properties’ accuracy depends on this parameter [39]. The frequency resolution was set to 0.101 Hz because it was compatible with the resonant frequencies and modal density estimated on the preliminary timber boards modal analysis. More details of the EFDD method can be found in [33].

SSI converts the second-order problem associated with the differential equation of motion into two first-order problems defined by the so-called state equation and observation equation. These equations contain both physical information and the vibration response of the structural element (system matrices) and statistical information of the dynamic input forces (vectors). In this method, the dynamic forces do not need to be measured, but they must comply with white-noise characteristics. The structural element’s modal parameters can then be extracted when both system matrices and vectors have been estimated. However, this task is quite challenging because SSI fits a parametric model directly to the raw structural vibration responses recorded in time series. The parametric model should have a reasonable number of parameters, also known as model order, to correctly represent both dynamic and statistical behavior. Nevertheless, since it is impossible to know the model order in advance, it is necessary to repeat the analysis for different model orders and verify the results’ repeatability through the so-called stabilization diagrams. More details of the SSI method can be found in [33].

To meet the requirements mentioned above, we decided to use 11 consecutive low-energy vertical impacts on the timber boards as a vibration excitation source. A rubber-tipped hammer was used to apply the impacts until the 60-s measurement time was completed. The impacts were applied at random locations on the top edge of the timber boards. These moving loads can be classified as multiple-input loads, a suitable alternative to satisfy the OMA assumptions of white noise excitations [40]. Besides, the set of 11 impacts was repeated three times on each timber board by different operators to evaluate the results’ repeatability.

In this study, only the dynamic properties associated with the first three resonant vibration modes in the vertical transverse direction were identified. This decision was based on numerical modal analysis of the timber boards carried out in previous studies [41]. In most numerical models, the first three resonant vibration modes had an equivalent accumulated modal mass slightly greater than 90% of the total mass. Therefore, these three vibration modes were sufficient to represent the timber boards’ vibration response. For the first, second, and third vibration modes, the effective modal mass was equivalent to 81%, 0%, and 9% of the total mass, respectively. Figure 6 shows the theoretical shapes of these three vibration modes.

The frequencies detected by EFDD and SSI methods will be compared through their percentage differences. In contrast, the modal shape vectors will be compared using the MAC (modal assurance criterion) indicator [42].

#### 2.3.2. Numerical Simulations

The timber boards’ geometry, density, and support conditions were replicated in numerical simulations applying the finite element method. The numerical models were implemented in the ANSYS^®^ software [43] using a 3-D two-node beam element called BEAM188, considering linear-elastic and orthotropic properties for wood. In typical engineering problems, nine elastic constants are needed to define an orthotropic material: Three elasticity modulus, three shear modulus, and three Poisson coefficients. However, in the present study, we only wanted to know the local variation of the modulus of elasticity in the longitudinal direction (*E_xdt_*); therefore, to reduce the number of unknown input parameters, a series of simplifications were made. Firstly, the recommendation of [44] was followed to approximate some elastic properties of hardwoods as a proportion of *E_xdt_* ($E_{y}\approx 0.14E_{xdt},\;E_{z}\approx 0.08E_{xdt},\;G_{xy}\approx 0.10E_{xdt},\;G_{yz}\approx 0.032E_{xdt},\;G_{xz}\approx 0.07E_{xdt}$). Secondly, a constant value of 0.35 was assumed for the three Poisson coefficients, as indicated in other investigations [4,5]. Finally, the local variation of *E_xdt_* was implemented by dividing the wood boards into 12 segments each of them with a length of 200 mm.

In each segment, *E_xdt_* could take a constant value within a predefined range. Thus, in each numerical simulation, 12 different *E_xdt_* values can be considered, distributed over the length of the timber boards. The segments’ length with constant properties was chosen to be like the maximum length of influence of a wood knot-type defect. According to the Chilean standard NCh1970-1 [45], that influence length is smaller than the double of the wood board height; therefore, 200 mm is a suitable segment length for this kind of hardwood timber boards. However, the segments are further divided into smaller parts to guarantee the proper discretization for applying the finite element method.

The variability range of the 12 input parameters was estimated from previous static bending tests carried out in Chile [46]. The experimental database used contained information from about 1000 *Eucalyptus nitens* boards from different plantations in Chile and with slightly different cross-sections than those of the present study. However, its results allowed establishing the ranges of minimum and maximum values for *E_xdt_* with a good level of agreement. Thus, it was defined for each model zone that *E_xdt_* could vary between 3 GPa and 20 GPa.

The number of numerical evaluations of the model (sample size) should be such that the variability range of the input parameters is well covered. According to [47], to study the model outputs with regional sensitivity analysis techniques, it is necessary to use sample sizes of at least 100 times the number of input parameters. Therefore, a sample size of 3000 was chosen, which amply exceeds the minimum value of 1200 recommended for this case. A Latin-Hypercube [48] sampling strategy was used since it covers better the space of variability of the input parameters than traditional random sampling. In addition, following the recommendations of [47], it was assumed that the input parameters were independent and uniformly distributed.

With the finite element model defined, modal analysis is carried out to obtain the theoretical dynamic properties of the first three vibration modes. Figure 7 shows a scheme that summarizes the numerical model details.

#### 2.3.3. Regional Sensitivity Analysis

Regional Sensitivity Analysis (RSA) is a set of methods that aim to identify the regions where a model’s input parameters cause extreme values in a specific output variable. RSA is especially useful for studying complex models with several input parameters of high variability. Often in such complex models, it is not easy to find optimal values for the input parameters, so priority is usually given to estimating their acceptable variation ranges. Another name for RSA is Monte Carlo filtering, initially proposed by [49,50] in environmental quality studies.

RSA implementation starts with defining an output variable *Y*, also called objective function. In our study, the *Y* function depends on the 12 values of the input parameters stored in the ***E****_xdtM_* vector. The *Y* function’s main feature is to contain the differences between the set of dynamic properties measured in the experimental tests and the set of the same properties calculated in the numerical models. According to the recommendations of [31], one of the suitable alternatives for defining the output variable Y is to use a normalized least-squares objective function, as shown in Equation (4).
(4)$$Y\left(E_{xdtM}\right)=\frac{1}{2}\cdot \overset{3}{\underset{r=1}{\sum }}a_{r}\cdot {\left(\frac{{\tilde{f}}_{r}^{2}-f_{r}^{2}\left(E_{xdtM}\right)}{{\tilde{f}}_{r}^{2}}\right)}^{2}+\frac{1}{2}\cdot \overset{3}{\underset{r=1}{\sum }}b_{r}\cdot {\left(\frac{\|\gamma_{r}\cdot {\tilde{\phi }}_{r}-L\cdot \phi_{r}{\left(E_{xdtM}\right)\|}_{2}}{\|\gamma_{r}\cdot {\tilde{\phi }}_{r}\|{}_{2}}\right)}^{2}\;\;\;,$$
where $a_{r}$ and $b_{r}$ are weighting factors, ${\tilde{f}}_{r}$ and $f_{r}\left(E_{xdtM}\right)$ are the measured and computed natural frequencies respectively, $\|\cdot {\|}_{2}$ is the Euclidean norm of a vector, ${\tilde{\phi }}_{r}$ and $\phi_{r}\left(E_{xdtM}\right)$ are the measured and computed mode shapes respectively, L is a binary matrix that selects the observed degrees of freedom from the degrees of freedom present in the finite element models, and $\gamma_{r}$ is a scaling factor to ensure that simulated and measured mode shapes are scaled equally. The analytical expression of $\gamma_{r}$ is shown in Equation (5).
(5)$$\gamma_{r}=\frac{{\tilde{\phi }}_{r}{}^{T}\;\cdot L\cdot \phi_{r}\left(E_{xdtM}\right)}{\|{\tilde{\phi }}_{r}{\|}_{2}^{2}},$$

After defining the *Y* function, it is necessary to evaluate it using the different sampled ***E****_xdtM_* values. In the RSA method, the *Y* values are calculated by varying all input parameters simultaneously. Therefore, each parameter’s sensitivity considers its direct influence and the joint influence due to the interactions between parameters. This sampling strategy is called the “All-[factors]-At-a-Time” method (AAT) and is characteristic of the Global Sensitivity Analysis (GSA), where the entire space of variability of the input parameters is studied.

Once the complete evaluation of the *Y* function is available, it is essential to divide the input parameters of the model into two binary sets, “behavioral” (B) and “non-behavioral” (NB), depending on whether the associated *Y* value is below or above a prescribed threshold value, *Y_t_*. As the Y function measures the normalized differences between the measured and calculated dynamic properties, the input parameters will belong to the B set if they generate Y values lower than *Y_t_*. In this way, a *Y_t_* value equal to the 5% percentile of the Y function values was assumed for each timber board. With that choice of *Y_t_*, it was assured that the B set would not generate weighted normalized differences between measured and calculated dynamic properties higher than 10% and 20% for the frequencies and vibration modes, respectively.

As soon as the input parameters have been divided in sets B and NB, their respective empirical cumulative distribution functions (CDF) must be calculated and plotted. Then, using the Kolgomorov–Smirnov (K-S) statistical test, it is evaluated whether there are statistically significant differences between the CDFs of sets B and NB. Finally, if the K-S test indicates significant differences, an appropriate distribution or transformation can be identified for the B parameter sets. This adjustment allows a more in-depth analysis of the ranges of variation of these moduli of elasticity and compare them with some normative limits. The computational implementation of the RSA method was done in the SAFE toolbox [51].

A crucial aspect for evaluating the objective function *Y* was selecting the *a_r_* and *b_r_* weighting factors shown in Equation (4). These weighting factors are six since, as discussed in Section 2.3.1, only the first three frequencies and modal forms were used. According to [31], these factors can be generally interpreted as a measure of confidence in the accuracy of the experimentally measured data; therefore, in most cases, these factors are estimated based on engineering criteria and trial-and-error.

A first approximation in the estimation was to assume that the weighting factors associated with the second resonant mode’s frequencies and modal shapes were equal to zero ($a_{2}=0$ and $b_{2}=0$). This decision was justified because the second vibration mode generally presents an effective modal mass that tends to zero, as shown in Figure 6. The second approximation was to assume that the weighting factors associated with the vibration modes’ errors were equal to half of their respective weighting factors associated with the frequencies’ errors ($b_{1}=0.5a_{1}$ and $b_{3}=0.5a_{3}$). The justification for this approximation is that the experimental frequencies are, in general, the most accurate experimental data, while the modes of vibration are more difficult to identify. So, it is common to apply lower weighting factors to the vibration modes [52]. With all the approximations made, the weighting factors to be determined were reduced to two ($a_{1}$ and $a_{3}$). These values were estimated iteratively until the mean value of the 12 sets of parameters B was equal to the $E_{xdt}$ value calculated with Equation (2), with a +/−1% error margin. Considering the 36 wood boards, the mean values obtained for the weighting factors a_1_ and a_3_ were 0.91 and 0.09, respectively. However, the general trend was that as the number of defects in a wood board increased, the value required for the a_3_ weighting factor increased. For example, in the timber board #33, which was one of the least defective, the weighting factors were a_1_ = 0.96 and a_3_ = 0.04. While in the timber board #19, which was one of the most defective, the weighting factors were a_1_ = 0.57 and a_3_ = 0.43.

## 3. Results and Discussion

### 3.1. Dynamic and Static Global Modulus of Elasticity

Table 2 shows a basic descriptive statistical summary of the *E_xdl_*, *E_xdt_*, and *E_x_* values, calculated with Equations (1)–(3), respectively. The results shown in Table 2 are interesting in several ways. First, it was observed that the mean value of the static modulus of elasticity (*E_x_*) was in a kind of central point between the mean values of both dynamic moduli of elasticity (*E_xdl_* and *E_xdt_*). When the vibrations were applied longitudinally, the mean value of *E_xdl_* was 8.7% higher than the mean value of *E_x_*, while when the vibrations were transverse, the mean value of *E_xdt_* was 8.9% lower than the mean value of *E_x_*. The abovementioned was confirmed through an analysis of variance (ANOVA) applied to the three moduli of elasticity. Based on the ANOVA results, it was concluded that there were differences among the means at the 0.05 level of significance (*p* = 0.005 < 0.05), especially between *E_xdl_* and *E_xdt_*. Second, the three experimental techniques showed coefficients of variation close to 20%, which is an acceptable level of dispersion in these test types. Finally, *E_x_*’s range of values is consistent with previous results of other investigations developed in *Eucalyptus nitens* timber boards from plantations in Chile [46] and Australia [11]. Unfortunately, in the investigations mentioned above, no dynamic modulus values were reported, which prevents extending the comparison to these non-destructive properties.

Figure 8 shows the regression analysis results between the dynamic and static elasticity modules (alpha level = 0.05). The relationship between the static and dynamic modulus of elasticity is statically significant (*p* < 0.001) and a positive correlation exists between them (r = 0.98). Besides, for both longitudinal (*E_xdl_*) and transversal (*E_xdt_*) vibration results, about 95% of the variation in E_x_ can be explained by the regression models. The fitted equations shown in Figure 8a,b can be used to predict E_x_ for a value of *E_xdl_* or *E_xdt_*, or find the settings for *E_xdl_* or *E_xdt_* that correspond to a desired value or range of values for *E_x_*. As far as we know, this is the first time that this kind of fitted equation has been developed for *Eucalyptus nitens* timber boards. However, both the correlation coefficients and R-squared values obtained are in good agreement with those reported in previous research on boards of different timber species [21,24,25,26].

### 3.2. Local Variation of Elasticity Modulus by Transverse Vibration

As mentioned in Section 2.3, the three-stage methodology for estimating local variation in *E_xdt_* combined operational modal analysis, numerical simulations, and regional sensitivity analysis. The most relevant results of these stages are shown below.

#### 3.2.1. Dynamic Properties of Lumber Boards

The first set of results required to obtain the wood boards’ dynamic properties was the measurement of their vibratory response to a series of low energy impacts. The vibration responses were essentially transient after each impact forces. Consequently, an initial peak response was followed by a free vibration of the timber boards that decayed due to the material damping ratio. Figure 9 shows the vibration response of the central accelerometer on timber boards #33. From the vibratory responses registered in the timber boards, their dynamic properties were estimated by the EFDD and SSI methods (Figure 10).

Figure 10 shows that, in general, the EFDD method estimated more directly the first three resonant frequencies. The above-mentioned is reflected in Figure 10a, where the light green areas highlighted in the singular value diagrams clearly show the three modal domains detected. Furthermore, in most of the timber boards, the detected frequencies coincided with the peaks of the singular-value line #1 (blue curve). Conversely, with the SSI method (Figure 10b), more resonant frequencies were detected (vertical-aligned red dots) in the same study range, some of which lacked physical sense. Therefore, the SSI method generated more post-processing work than the EFDD method. However, after making the necessary adjustments with the two methods, very similar dynamic properties were obtained. Some comparative results of the frequencies and modes of vibration obtained with the EFDD and SSI methods in different timber boards are shown in Table 3 and Table 4. Three criteria were used to select the cases shown in these Tables: (i) To consider at least 25% of the total cases, (ii) to include cases that represent the whole range of variation of *E_x_*, and (iii) to have a mean value of E_x_ that does not differ by more than 5% from that reported in Table 2.

From the results shown in Table 3, several relevant aspects can be observed. First, it is evident that for timber boards of different structural quality, the estimated mean values of the first resonant frequency with the EFDD and SSI methods are similar, with differences smaller than 2%. Second, this similarity between the results of the methods is also observed in estimating the first mode of vibration. When comparing the modal vectors of the EFDD method concerning the SSI method, MAC indicators close to 1 were obtained. Third, the coefficients of variation of the frequencies, product of each the three test repetitions, were low for both methods, with values smaller than 1%. All the above suggests that both the values of the first resonant frequency and its respective modal form are quite reliable. There were no statistically significant differences between the two methods that have a very different theoretical framework.

Most of the previous paragraph comments are also applicable to the experimental results obtained for the third resonant frequency and its respective mode shape shown in Table 4. Although the differences between the third resonant frequencies obtained by the EFDD and SSI methods increased compared to the first resonant frequency, they remained reasonably low, not exceeding 3%. However, the coefficients of variation of the third resonant frequency were higher than those obtained in the first resonant frequency, especially in the SSI method. The above shows that both methods had a little more difficulty for detecting the third resonant frequency in this experimental configuration of simply supported timber boards. Finally, the MAC indicator between the EFDD and SSI modal vectors also suffered a minor decrease; however, it always was higher than 0.97. Figure 11 shows the graphical representations of the first and third modal shapes for the wood boards with the best and worst MAC indicators.

#### 3.2.2. Regional Sensitivity Analysis of *E_xdt_* Local Variation

Once the timber boards’ experimental dynamic properties were reliably estimated, the numerical simulations were carried out, as commented in Section 2.3.2. Thus, having both the set of measured and calculated dynamic properties, the objective function indicated in Equation (4) was evaluated. Then, the output threshold value Y_t_ was defined for each timber board, and the input parameters sampled were separated in the sets B and NB. The CDFs of sets B and NB were calculated with this information, and the K-S test was applied. Finally, it was studied how *E_xdt_* varied in sets B along the timber boards.

According to the Chilean structural design regulations for timber constructions [44], if the studied *Eucalyptus nitens* boards were to be used as beams in social housing lightweight floor systems, they should not have vertical displacements greater than 6.7 mm when subjected to live loads of 2 kN/m^2^. The above mentioned implies that each beam must have an *E_x_* value equal to or greater than 10.35 GPa to comply with the serviceability restrictions. If this normative limit wants to be expressed in terms of the *E_xdt_* parameter, the adjustment equation shown in Figure 8b can be used. In this way, it was obtained that *E_xdt_* should be greater than or equal to 9.4 GPa. Figure 12, Figure 13, Figure 14, Figure 15, Figure 16, Figure 17, Figure 18, Figure 19 and Figure 20 show the most relevant results of the RSA for the local variation of *E_xdt_* for three cases of timber boards with different structural qualities. For timber beams that are simply supported and subjected to vertical loads, the areas subjected to the most significant vertical displacement are those corresponding to the central third of their length. Therefore, Figure 12, Figure 13, Figure 14, Figure 15, Figure 16, Figure 17, Figure 18, Figure 19 and Figure 20 only show the results in zones 5, 6, 7, and 8, defined in Figure 7.

The scatter plots in Figure 12, Figure 15, and Figure 18 allowed visualizing the general distribution of *E_xdt_* values in sets B and NB for different structural quality timber boards. As reference, wood boards #33, #19, and #28 had *E_xdt_* values evaluated with Equation (2) of 12.79 GPa, 10.54 Gpa, and 9.68 Gpa, respectively. These differences in global dynamic elasticity moduli were also confirmed at the local level. Broadly speaking, as the structural quality of the timber boards decreased, the degree of dispersion of the *E_xdt_* values in the B sets for zones 5 to 8 increases. For example, while in timber board #33, the minimum values of *E_xdt_* in the B sets were around 9 Gpa (Figure 12); in timber boars #19 and #28, those minimum values were around 6 Gpa (Figure 15) and 4 Gpa (Figure 18), respectively.

The CDF plots in Figure 13, Figure 16, and Figure 19 allowed analyzing the influence of the *E_xdt_* values on the objective function Y for timber boards of different structural quality. According to the RSA method, the influence of an input parameter (*E_xdt_*) on the output variable (Y) can be estimated from the K-S test applied to the CDFs in sets B and NB. Thus, the lower the p-value of the K-S test, the more influential the input parameter (*E_xdt_*) is on the output variable (Y). Therefore, some authors [53] suggest that the input parameters (*E_xdt_*) can be grouped into three sensitivity classes: Critical (*p*-value < 0.01), important (0.01 < *p*-value < 0.1), and negligible (*p*-value > 0.1). The results shown in Figure 13, Figure 16, and Figure 19 indicated that as the timber boards’ structural quality decreased, their respective *E_xdt_* parameters were less influential on the Y function. The above was also evident in the greater difficulty in differentiating the CDFs from the B sets concerning the NB sets for timber boards of lower structural quality. For example, in timber boards #33, #19, and #28, the *E_xdt_* parameters of zones 5 to 8 were classified on average as critical (*p*-value = 1 × 10^−13^), critical (*p*-value=2 × 10^−5^), and important (*p*-value = 4 × 10^−2^), respectively.

Finally, the boxplots in Figure 14, Figure 17, and Figure 20 allowed studying the variability of *E_xdt_* in the B sets. It is convenient to remember that the B sets correspond to those numerical models whose input parameters *E_xdt_* generated dynamic properties closest to the experimental measurements. Therefore, knowing the *E_xdt_* variability in B sets is very useful when the normative serviceability limits want to be checked from a probabilistic approach. Through the quality tools of Minitab software [54], it was identified that the distribution that best fits the *E_xdt_* values in the B sets was the Johnson transformation [55]. With these adjustments, it was possible to estimate in each zone of *E_xdt_* the probability of having a value lower than the serviceability limit of 9.4 GPa. For example, in the timber boards #33, #19, and #28, the lowest probability of having an *E_xdt_* value lower than 9.4 GPa was 9.8% (zone 6), 22.53% (zone 5), and 44.9% (zone 7), respectively.

The results mentioned in the previous paragraph are quite interesting since they allow the definition of specific structural serviceability criteria for this type of timber. If only a traditional structural serviceability criterion based on the global *E_xdt_* evaluation with Equation (2) were considered, the three analyzed timber boards would comply satisfactorily since their *E_xdt_* values are higher than 9.4 GPa. Considering the global results mentioned above, only the timber board #33 fits well with the photographic records (Figure 14) since the entire central third of the board is free of defects. However, when observing the size and distribution of knots and cracks in the central third of timber boards #19 (Figure 17) and #28 (Figure 20), serious doubts arise about that criterion’s effectiveness. Therefore, it is necessary to perform a deeper analysis of the local variability of *E_xdt_*.

On the one hand, the timber board #19 had three knots in its central third with sizes greater than 3/8 of its height, which is the limit recommended by the Chilean standard for hardwoods [45]. For this reason, the probabilities of having *E_xdt_* values lower than 9.4 GPa increased in this wood table until reaching 22.53%, 21.37%, 20.64%, and 20.52% for zones 5, 6, 7, and 8, respectively. The similarity between the probabilities in these four zones was because the knots were almost uniformly distributed and had similar sizes. However, the method used was able to assign a slightly higher probability in zone 5, where the largest knot was located. The probabilities were not higher because the knots did not affect both sides of the timber board.

On the other hand, timber board #28 had a combination of knots and horizontal cracks in its central third, which appeared on both sides. The defects mentioned caused a significant increase in the probabilities of having values below 9.4 GPa, with 29.00%, 34.82%, 44.9%, and 29.44% for zones 5, 6, 7, and 8, respectively. The higher probability assigned by the method for zone 7 is fully justified since, in that sector, the horizontal crack moves closer to the bottom edge and crosses to both sides of the timber board. Therefore, in a structural serviceability check, it seems more reasonable to use methods that study local *E_xdt_* variability than methods that use only global *E_xdt_* values.

By extending the serviceability checks to the rest of the timber boards, two clear trends are obtained. First, if the current normative criterion is used that the global *E_xdt_* value must exceed 9.4 GPa, 23 wood boards would comply (64% of the total). However, 8 of the 23 accepted boards had relevant defects in their central thirds, questioning the acceptance criterion’s validity. Second, if a local criterion is applied, such that all *E_xdt_* in the central third must have a probability of not exceeding 9.4 GPa of less than 20%, only 15 wood boards would comply (42% of the total). What is interesting about this new criterion is that the eight wood boards, whose compliance with the current global criterion was being questioned because of their defects, would now be placed in the group of those who do not comply. Figure 21 shows the central third of the eight wood boards mentioned above. Therefore, this new local criterion has a more significant potential to be positively correlated with visual assessments of defects in *Eucalyptus nitens* timber boards.

Finally, it is essential to mention that the approximations made in the definition of the nine elastic properties of the wood in the numerical models were quite adequate (Section 2.3.2). Firstly, these approaches satisfy the relationships that must exist between the modulus of elasticity and the Poisson coefficients in orthotropic materials $\left(v_{ij}\leq \sqrt{E_{i}/E_{j}}\right)$. Besides, the stress-strain matrix is always positive definite when using the proposed values for the nine orthotropic elastic constants. Secondly, the elasticity modulus in the timber boards’ longitudinal direction (*E_x_*) was the most influential variable in estimating its dynamic properties. Of the rest of the elastic properties included in the numerical model, the second most influential was the shear modulus *G_xy_*. For example, a change in the *G_xy_* magnitude of 20% caused variations in the frequencies *f_1_* and *f_3_* of 0.3% and 2.2%, respectively. These small variations were because the timber boards evaluated had a length/height ratio greater than 20; therefore, the boards’ dynamic response is dominated by bending and the shear deformations were negligible. The rest of the elastic properties had much less influence than *G_xy_*, so it was adequate to leave them expressed as a fraction of *E_x_* or as constants. This decision allowed us to simplify the dynamic problem without losing precision in the results significantly. However, these approaches’ effectiveness should be studied in more depth for timber boards with length/height ratios of less than 20.

## 4. Conclusions

This paper has highlighted the importance of the non-destructive assessment of *Eucalyptus nitens* timber boards’ bending stiffness. The experimental campaign revealed ranges of static and dynamic modulus of elasticity for this kind of timber boards by applying static forces, longitudinal vibrations, and transverse vibrations. The defects present in the timber boards, together with the combination of operational modal analysis, numerical simulations, and regional sensitivity analysis, have allowed the current knowledge about this research topic to be extended.

The positive correlation found between the global static and dynamic modulus of elasticity suggests that the vibration-based non-destructive techniques could be used as a reliable alternative for bending stiffness assessment of *Eucalyptus nitens* timber boards with different kinds of defects.

The results obtained in verifying the timber boards’ serviceability against vertical displacements suggest that the dynamic elasticity modulus’s local variability could be a better stiffness-based quality indicator than the traditional global elasticity modulus. For example, a good indicator was the probability of not exceeding a dynamic elasticity modulus normative limit in the timber boards’ central third. By requiring that this probability be less than 20%, the results fitted very well with the presence of knots and cracks shown in the timber boards’ photographic records. Therefore, timber boards with few or none defects in its central thirds met the serviceability criterion while those with relevant defects did not.

Finally, the results of this study support the idea that *Eucalyptus nitens* timber boards could be used as structural elements promoting more sustainable constructions. However, since it is common for these woods to present a significant number of defects, it is essential to select only those with high bending stiffness to avoid future complains by users. To overcome these challenges, the non-destructive evaluation methods presented in this study proved to be very useful. Future work will investigate *Eucalyptus nitens* timber boards of different structural dimensions.

## Figures and Tables

**Figure 1 materials-14-00269-f001:**
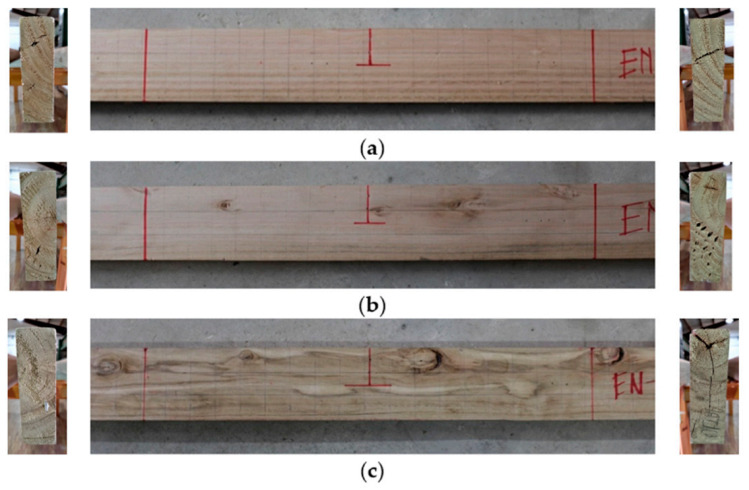
Lateral view of the central third and transversal section of: (**a**) Timber board #33, (**b**) timber board #28, and (**c**) timber board # 10.

**Figure 2 materials-14-00269-f002:**
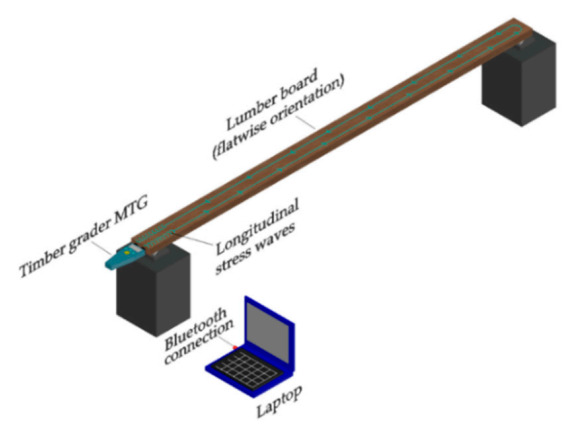
Longitudinal vibration test set up.

**Figure 3 materials-14-00269-f003:**
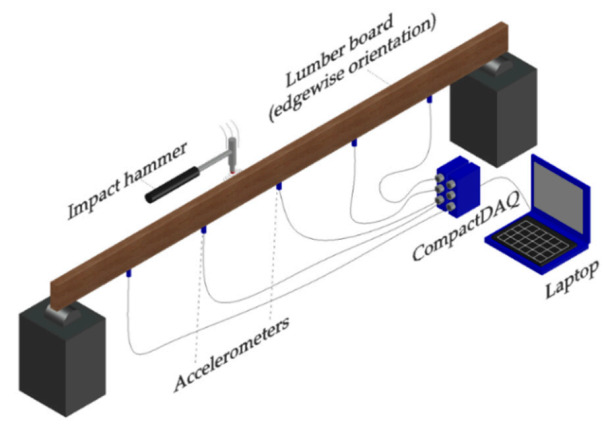
Transverse vibration test set up.

**Figure 4 materials-14-00269-f004:**
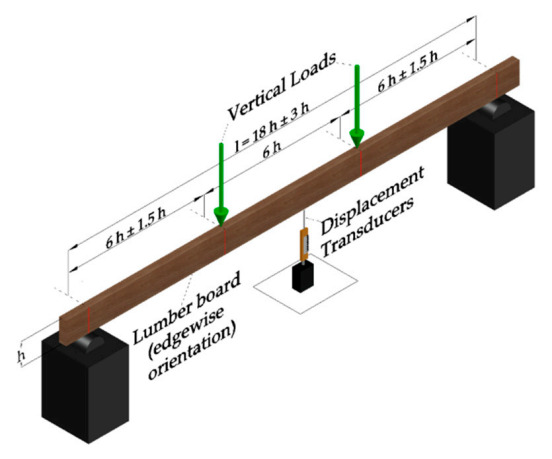
Static bending test set up.

**Figure 5 materials-14-00269-f005:**
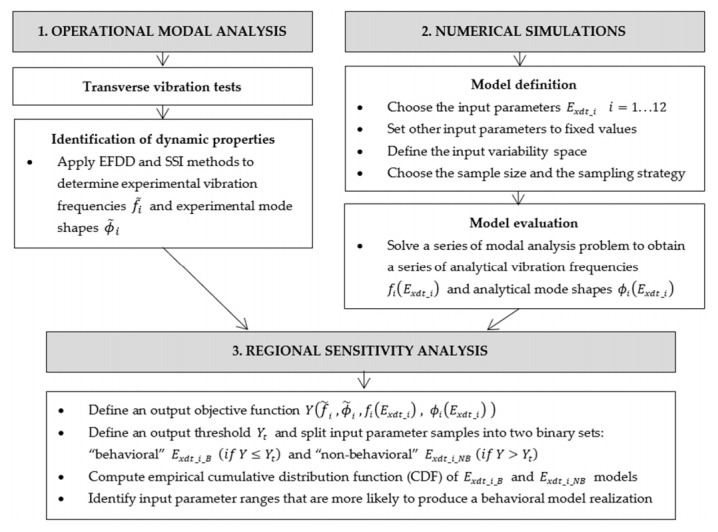
Logical diagram of *E_xdt_* local variability estimation.

**Figure 6 materials-14-00269-f006:**
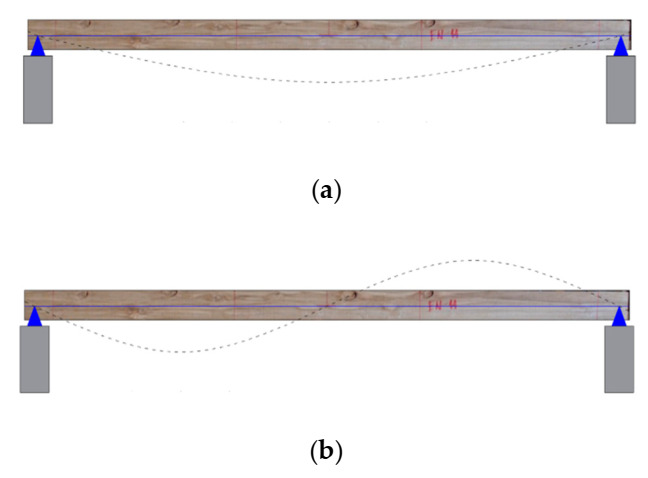
Theoretical first three resonant vertical vibration modes: (**a**) First mode, (**b**) second mode, and (**c**) third mode.

**Figure 7 materials-14-00269-f007:**
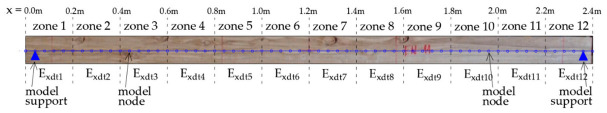
Numerical model of timber boards.

**Figure 8 materials-14-00269-f008:**
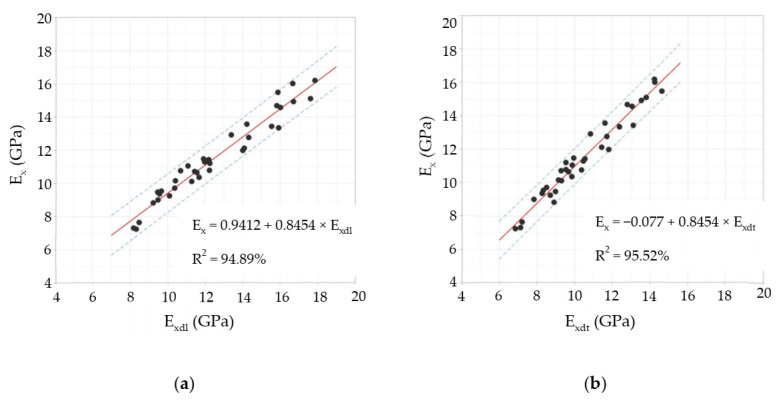
Regression analysis for: (**a**) *E_x_* versus *E_xdl_*; (**b**) *E_x_* versus *E_xdt_*. The red fitted lines show the predicted *E_x_* for any *E_xdl_* or *E_xdt_* value. The blue dashes lines show the 95% prediction interval.

**Figure 9 materials-14-00269-f009:**
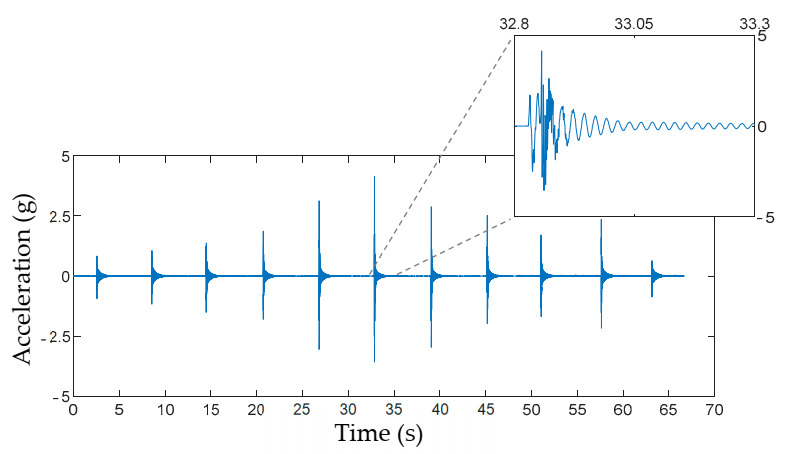
Vibrational response for the timber board #33 measured in the central accelerometer.

**Figure 10 materials-14-00269-f010:**
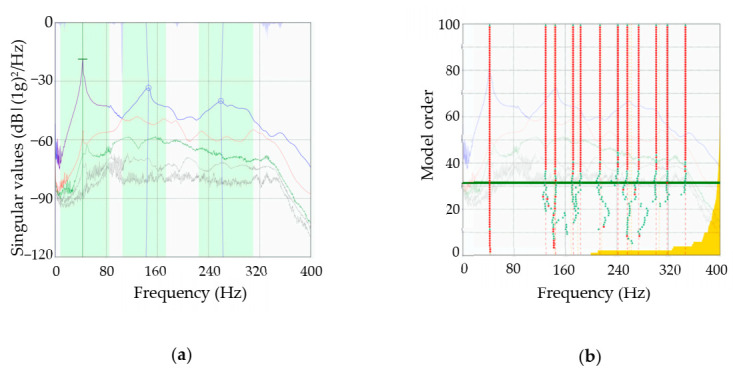
(**a**) Singular values diagram for lumber board #33 (Enhanced Frequency Domain Decomposition (EFDD) method); (**b**) stabilization diagram for lumber board #33 (Stochastic Subspace Identification (SSI) method). Adapted from ARTeMIS Modal Pro [38].

**Figure 11 materials-14-00269-f011:**
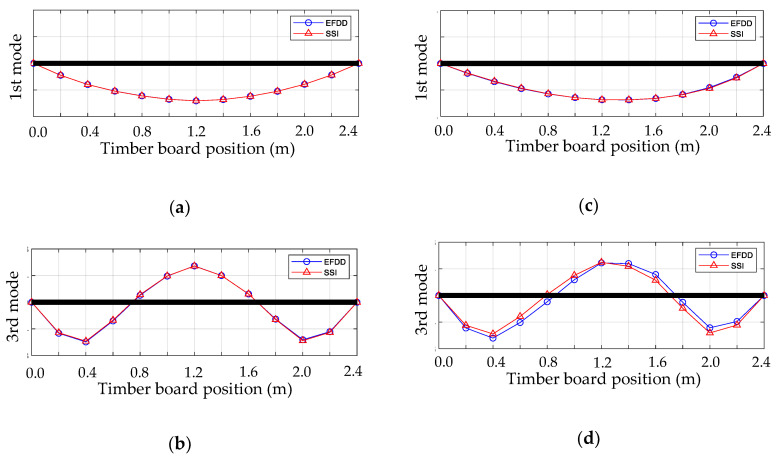
Graphical comparison between the experimental modal shapes obtained by EFDD and SSI methods in: (**a**) Timber board #20—first mode; (**b**) timber board #20—third mode; (**c**) timber board #28—first mode; (**d**) timber board #28—third mode.

**Figure 12 materials-14-00269-f012:**
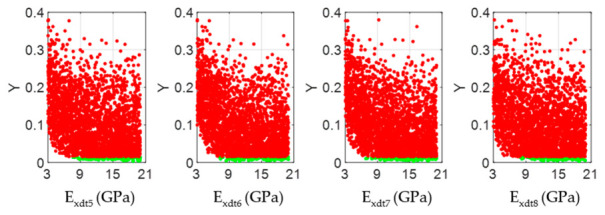
Scatter plots of the output samples (Y function) against the four input factors located in the central third of the timber board # 33 (*E_xdt_* in zones 5 to 8). The green and red dots represent the sets B (behavioral) and NB (non-behavioral), respectively.

**Figure 13 materials-14-00269-f013:**
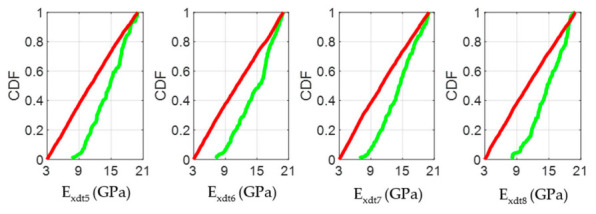
Cumulative Distributions Functions (CDFs) of the four input factors located in the central third of the timber board # 33 (*E_xdt_* in zones 5 to 8) for the output metric (Y function). The green and red curves represent the sets B (behavioral) and NB (non-behavioral), respectively.

**Figure 14 materials-14-00269-f014:**
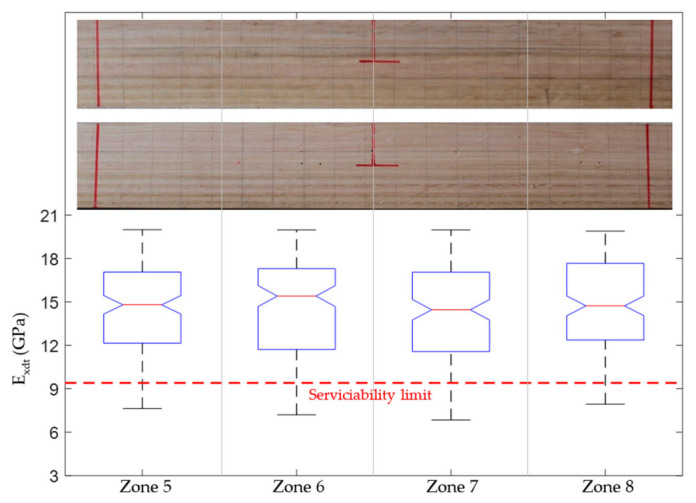
Boxplots of the four input factors located in the central third of the timber board # 33 (*E_xdt_* in zones 5 to 8) for B sets. The pictures show the back and front of the lumber board in zones 5 to 8.

**Figure 15 materials-14-00269-f015:**
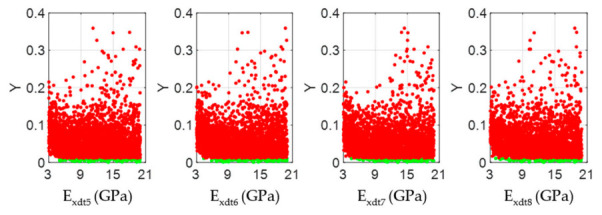
Scatter plots of the output samples (Y function) against the four input factors located in the central third of the timber board # 19 (*E_xdt_* in zones 5 to 8). The green and red dots represent the sets B (behavioral) and NB (non-behavioral), respectively.

**Figure 16 materials-14-00269-f016:**
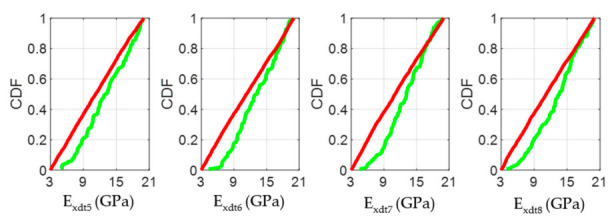
Cumulative Distributions Functions (CDFs) of the four input factors located in the central third of the timber board # 19 (*E_xdt_* in zones 5 to 8) for the output metric (Y function). The green and red curves represent the sets B (behavioral) and NB (non-behavioral), respectively.

**Figure 17 materials-14-00269-f017:**
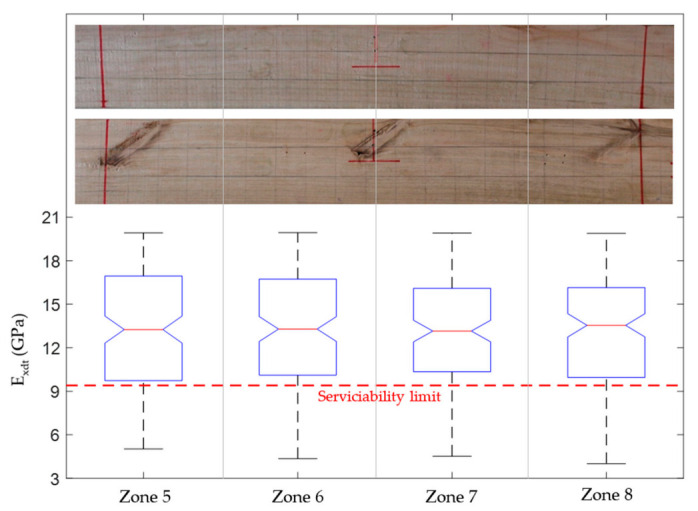
Boxplots of the four input factors located in the central third of the lumber board # 19 (*E_xdt_* in zones 5 to 8) for B sets. The pictures show the back and front of the timber board in zones 5 to 8.

**Figure 18 materials-14-00269-f018:**
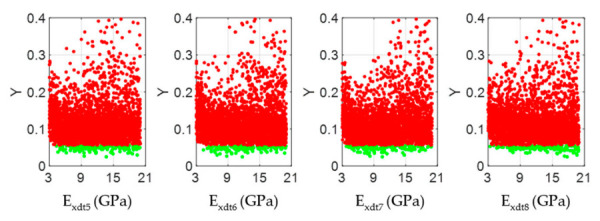
Scatter plots of the output samples (Y function) against the four input factors located in the central third of the timber board # 28 (*E_xdt_* in zones 5 to 8). The green and red dots represent the sets B (behavioral) and NB (non-behavioral), respectively.

**Figure 19 materials-14-00269-f019:**
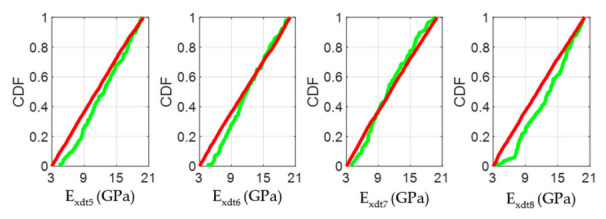
Cumulative Distributions Functions (CDFs) of the four input factors located in the central third of the timber board # 28 (*E_xdt_* in zones 5 to 8) for the output metric (Y function). The green and red curves represent the sets B (behavioral) and NB (non-behavioral), respectively.

**Figure 20 materials-14-00269-f020:**
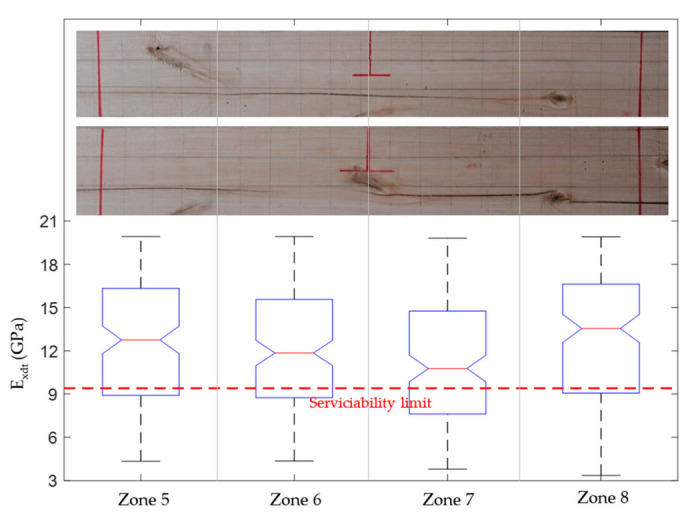
Boxplots of the four input factors located in the central third of the timber board # 28 (*E_xdt_* in zones 5 to 8) for B sets. The pictures show the back and front of the lumber board in zones 5 to 8.

**Figure 21 materials-14-00269-f021:**
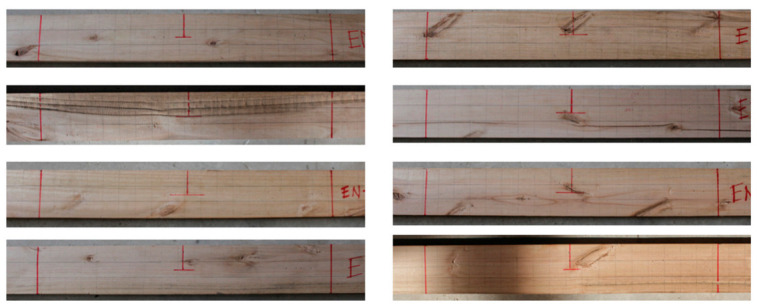
Defect visualization in the central third of the eight timber boards that met the current global serviceability criterion but not the proposed new local criterion.

**Table 1 materials-14-00269-t001:** Dimensions, visual characteristics, and physical properties of timber boards.

-	Thickness (mm)	Height (mm)	Length (mm)	Continuous Clear Wood Length (mm)	Moisture Content (%)	Density (kg/m^3^)
Minimum	33.83	116.17	2413.00	120.84	11.93	473.81
Maximum	37.83	120.83	2422.30	2295.87	17.77	697.82
Mean	36.65	118.11	2416.70	978.04	14.42	571.04
SD	0.63	0.87	3.14	584.48	1.42	58.53
COV: %	1.72	0.74	0.13	59.76	9.86	10.25

**Table 2 materials-14-00269-t002:** Global mechanical properties for the *Eucalyptus nitens* wood sample.

-	$E_{xdl}\;(\mathrm{GPa})$	$E_{xdt}\;(\mathrm{GPa})$	$E_{x}\;(\mathrm{GPa})$
Minimum	8.19	6.84	7.23
Maximum	17.85	14.62	16.18
Mean	12.50	10.48	11.50
SD	2.83	2.17	2.45
COV: %	22.65	20.72	21.33

**Table 3 materials-14-00269-t003:** Comparison of first resonant frequency and mode shape obtained with EFDD and SSI methods in different timber boards.

Board #	$E_{x}\;(\mathrm{GPa})$	Density (kg/m^3^)	EFDD	SSI	$\frac{f_{1}^{SSI}-f_{1}^{EFDD}}{f_{1}^{EFDD}}$	MAC_1_
			*f*_1_ (Hz)	*f*_1_ (Hz)		
33	14.67	621.87	42.77 (0.16%)	42.78 (0.16%)	+0.03%	0.99998
20	13.35	648.95	41.28 (0.21%)	40.93 (0.38%)	−0.87%	1.00000
30	12.11	579.51	42.13 (0.37%)	41.90 (0.25%)	−0.55%	0.99996
19	11.41	609.25	39.65 (0.10%)	39.55 (0.05%)	−0.25%	1.00000
17	11.05	582.82	39.19 (0.19%)	38.65 (0.28%)	−1.38%	0.99998
28	10.65	565.68	39.34 (0.59%)	38.89 (0.52%)	−1.15%	0.99980
15	9.53	508.25	37.91 (0.69%)	37.49 (0.72%)	−1.10%	0.99999
29	8.81	529.05	39.00 (0.12%)	38.48 (0.35%)	−1.33%	0.99986
10	7.63	569.74	33.37 (0.20%)	32.80 (0.21%)	−1.71%	0.99994

Note: The coefficients of variation of the frequencies are shown in parenthesis.

**Table 4 materials-14-00269-t004:** Comparison of third resonant frequency and modal shape obtained with EFDD and SSI methods in different timber boards.

Board #	$E_{x}\;(\mathrm{GPa})$	Density (kg/m^3^)	EFDD	SSI	$\frac{f_{3}^{SSI}-f_{3}^{EFDD}}{f_{3}^{EFDD}}$	MAC_3_
			*f*_3_ (Hz)	*f*_3_ (Hz)		
33	14.67	621.87	263.29 (1.38%)	257.83 (0.55%)	−2.07%	0.99667
20	13.35	648.95	244.10 (0.89%)	246.20 (0.62%)	+0.86%	0.99960
30	12.11	579.51	222.75 (1.38%)	225.13 (5.72%)	+1.07%	0.99699
19	11.41	609.25	278.31 (1.18%)	279.55 (1.20%)	+0.45%	0.99880
17	11.05	582.82	200.40 (0.81%)	194.77 (1.91%)	−2.81%	0.99587
28	10.65	565.68	241.55 (2.60%)	234.33 (11.9%)	−2.99%	0.97064
15	9.53	508.25	233.73 (1.82%)	239.69 (0.35%)	+2.55%	0.99804
29	8.81	529.05	244.30 (0.54%)	241.41 (0.14%)	−1.18%	0.98261
10	7.63	569.74	211.63 (0.75%)	207.92 (0.75%)	−1.75%	0.99458

Note: The coefficients of variation of the frequencies are shown in parenthesis.

## Data Availability

The data presented in this study are available on request from the corresponding author. The data are not publicly available due to privacy restrictions.

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
