# Peer review of "Non-Destructive Assessment of the Dynamic Elasticity Modulus of Eucalyptus nitens Timber Boards"

_materials, 2021, doi:10.3390/ma14020269_

Round 1

Reviewer 1 Report

In general, the paper looks good and ready for publication after some minor changes.
Nevertheless, the structure and overall quality can be improved. Paragraphs with +20 lines seem excessive and wordy. The English style should also be improved (e.g the expression “on the other hand” is used 11 times, and while it can be fine, it is something that called my attention).

The materials and methods section could be better structured and shorter as 8 pages seems a bit excessive.

Data (eg Table 1) needs to have errors or SD values.

I cannot comment on the significance of the results, as this is not my area.

Reviewer 2 Report

This manuscript presents non-destructive testing to determine the dynamic modulus of Eucalyptus timber boards using the standard vibration test. In my opinion this is a well written paper with all results presented well. I think this manuscript is almost ready for publication and I do not have any major questions for authors. 

Reviewer 3 Report

The aim of the manuscript is to evaluate the dynamic elasticity modulus of Eucalyptus nintens timber boards through non-destructive vibration-based tests. These technic are interesting both from a practice and science point of view. The manuscript is well written and designed.

The goal of the paper is clear. The title is clear and reflects the study content. The abstract contains the necessary information. Materials and methods are described very well. The authors have provided all necessary information for a potential reader, less familiar with the respective field, demonstrating a good overview of the problem. However, I think that the details of the methodology should be moved to the annex. For example, chapters 2.2 and 2.3. These chapters are very interesting, but I suggest you give only the relevant details.

Specific items:

L73-74: Literatures should be included.

L95-97: The same sentences as ones found in lines 42-44; these need to be rewritten.

L97-102: This is a methodology. I propose to highlight this information at the beginning of Chapter 2.

L114-115: Why representative, not an example? The authors do not provide any criteria for selecting timber boards. What part of the tree trunk were the boards taken from? This is important for wood density and other properties.

L120-121: How long have the boards were air-conditioned? Relative humidity (75%) is correct? I do not understand why the moisture content was again measured before the test.

Table 1. Please explain in the text what is "Continous Clear Wood Length".

L150-152: Move sentence to line 145.

Table 2. Were the means statistically different?

Figure 8: Correct R2.

L390: Statistical procedures not described? Please add explanations.

L404-410: This is not a description of the results. Move sentences to materials and methods.

Table 3 and Table 4: Please provide an explanation, why in tables was shown only selected results. Why only selected timber boards? What method of selection was used? The wooden planks have not been thoroughly characterized. Not enough information about materials (see chapter 2.1.).

The conclusions are consistent with the results.

References are well chosen. They indicate the authors' overview of the respective problem. References are nowel, as well as older, to cover the development of the understanding of the respective problem.

Concluding, I recommend publishing the paper after successful minor revisions.

Reviewer 4 Report

The article presents an interesting study regarding the estimation of the modulus of elasticity parallel to the grain in Eucalyptus nitens timber boards through non- destructive testing based on the vibration response. The novelty of the paper is that this estimation is performed not only globally, as in previous works, but locally along the structural element.  The local variability of the dynamic elasticity modulus should be a superior stiffness-based quality indicator than the traditional global elasticity modulus since it takes into account different kinds of local defects in the timber boards.

The article is well written and can be of significance in the field of the sustainable construction industry. However, before publication, some clarifications and improvements are needed:

  • In the study, the authors state that, in order to calculate the local variation of Exdt, several vibration modes are needed. For this purpose, two different OMA techniques are used to obtain the vibration modes. However, the authors initially calculate the dynamic properties associated only with the first three resonant vibration modes in the vertical transverse direction, discarding the higher modes; at the end, they use just the first mode of vibration in the calculation of the objective function (equation 4), rejecting the second mode and applying a very small weighting factor to the dynamic properties corresponding to the third mode that makes this mode negligible. Correspondingly, in this reviewer’s opinion, the use of both OMA techniques seems to be redundant. Why did the authors not consider more modes in the calculation of the objective function? Moreover, if only the first mode is used, the objective function (eq.4)  can be simplified.
  • Furthermore, in the expression of the objective function (eq. 4), what is the point of applying the coefficient ½ to both terms in the equations?
  • Since timber is an orthotropic material, nine elastic constants are needed in the finite element model. Some hypotheses where made in the model: ??≈0.14????, ??≈ 0.08????, ???≈0.10????, ???≈0.032????, ???≈0.07???? and the three Poisson coefficients equal to 0.35. A discussion should be made at the end on how these assumptions may influence the final results.
